# *Pseudomonas aeruginosa* Infections in Cancer Patients

**DOI:** 10.3390/pathogens11060679

**Published:** 2022-06-12

**Authors:** Paulina Paprocka, Bonita Durnaś, Angelika Mańkowska, Grzegorz Król, Tomasz Wollny, Robert Bucki

**Affiliations:** 1Department of Microbiology and Immunology, Institute of Medical Science, Collegium Medicum, Jan Kochanowski University, IX Wieków Kielc 19A, 25-317 Kielce, Poland; paulina.paprocka@ujk.edu.pl (P.P.); bonita.durnas@ujk.edu.pl (B.D.); angelika.mankowska@ujk.edu.pl (A.M.); grzegorz.krol@ujk.edu.pl (G.K.); 2Holy Cross Oncology Center of Kielce, Artwińskiego 3, 25-734 Kielce, Poland; tomasz.wollny@onkol.kielce.pl; 3Department of Medical Microbiology and Nanobiomedical Engineering, Medical University of Białystok, Jana Kilińśkiego 1 Białystok, 15-089 Białystok, Poland

**Keywords:** *Pseudomonas aeruginosa*, cancer patients, infections, new antibiotics

## Abstract

*P**seudomonas aeruginosa* (*P. aeruginosa*) is one of the most frequent opportunistic microorganisms causing infections in oncological patients, especially those with neutropenia. Through its ability to adapt to difficult environmental conditions and high intrinsic resistance to antibiotics, it successfully adapts and survives in the hospital environment, causing sporadic infections and outbreaks. It produces a variety of virulence factors that damage host cells, evade host immune responses, and permit colonization and infections of hospitalized patients, who usually develop blood stream, respiratory, urinary tract and skin infections. The wide intrinsic and the increasing acquired resistance of *P. aeruginosa* to antibiotics make the treatment of infections caused by this microorganism a growing challenge. Although novel antibiotics expand the arsenal of antipseudomonal drugs, they do not show activity against all strains, e.g., MBL (metalo-β-lactamase) producers. Moreover, resistance to novel antibiotics has already emerged. Consequently, preventive methods such as limiting the transmission of resistant strains, active surveillance screening for MDR (multidrug-resistant) strains colonization, microbiological diagnostics, antimicrobial stewardship and antibiotic prophylaxis are of particular importance in cancer patients. Unfortunately, surveillance screening in the case of *P. aeruginosa* is not highly effective, and a fluoroquinolone prophylaxis in the era of increasing resistance to antibiotics is controversial.

## 1. Introduction

Anticancer chemotherapy, radiotherapy, surgery, stem cell transplantation, and immunotherapy as well as cancer itself make oncological patients vulnerable to infection, which is the second cause of death in this population. The important risk factors of infections are neutropenia and disruption of anatomic barriers such as skin and mucous membranes [1]. One of the most challenging aspects of patient management is the treatment of infections caused by multidrug-resistant (MDR) Gram-negative bacteria, e.g., carbapenemase-producing *Enterobacterales* and resistant non-fermentative rods. Overall, the emergence of antibiotic resistance reduces the effectiveness of antibiotic therapy, increases infection mortality, and significantly interferes with anti-cancer therapy [2].

This review presents selected aspects of infections in patients with solid tumors and hematological malignances caused by the most frequent non-fermentative, Gram-negative rod *P. aeruginosa*. We focus on microbial pathogenic factors, the clinical form of infections, the mechanisms of antibiotic resistance, available treatment options, and the control of nosocomial infections.

## 2. Features of Microorganism, Virulence Determinants

*P. aeruginosa* is a Gram-negative opportunistic pathogen, which has an outstanding adaptive capability, and thus is widely distributed in various habitats such as soil, river, artificial water reservoirs, the hospital environment and the human body [3]. This bacterium can affect not only humans (mainly immunocompromised patients, e.g., suffering from cancer, AIDS, burns) but also plants, pets and farm animals. The pathogenicity of *P. aeruginosa* is the result of the high plasticity of the genome, the production of numerous virulence factors, the ability of biofilm formation and the resistance to different classes of antibiotics. These features are regulated by nucleotide signals and the quorum sensing system [4]. The *P. aeruginosa* features influencing pathogenicity are presented in Figure 1.

*P. aeruginosa* can produce an extensive number of cell-associated and extracellular virulence factors, e.g., LPS (lipopolysaccharide), alginate, adhesins, pili and flagella, siderophores (pyoverdine and pyochelin), lipases, proteases, elastases, exotoxin A, and pyocyanin, as well as secretion systems releasing effector substances–toxins and hydrolytic enzymes. The virulence factors are involved in the various stages of the infection: adherence to the biotic and abiotic surfaces, colonization, dissemination and tissue damage [5]. The secretion system releases the compounds into the extracellular milieu or into the cells. For example, the T3SS (type III secretion system), variably expressed in different strains, can inject toxins such as ExoU, ExoT, ExoS, and ExoY directly into host cells, causing cytotoxicity and damage to the host’s immune system. The major toxin ExoU is associated with severe acute lung infections, sepsis and increased risk of early mortality [6]. The type III secretion system (T3SS) is important in the pathogenesis of invasive and acute infections, but most strains of *P. aeruginosa* that cause chronic infections/colonization (e.g., in cystic fibrosis patients) may inhibit or even lose this and some other virulence factors, which result in immune evasion and host adaptation [7].

T2SS secrets exotoxin A (PE), the most toxic virulence factor produced by the majority of *P. aeruginosa* clinical strains. It is released into the extracellular environment and displays local necrotic activity [8]. The high cytotoxic properties of *Pseudomonas* exotoxin A are used to develop recombinant immunotoxins for anti-cancer therapy. The enzymatic extoxin A domain is linked to antibody fragments that target a specific antigen on tumor cells. Upon entry into the cell, the toxin domain inactivates protein biosynthesis and induces the process of apoptosis. In 2018, Moxetumomab Pasudotox (Lumoxiti), the drug for the treatment of drug-resistant hairy cell leukemia, was approved by the Food and Drug Administration. Research is underway on the development of new PE-based immunotoxins with potential anti-cancer effects on solid tumors [9]. The last type VI secretion system (T6SS) present in *P. aeruginosa* strains facilitates the colonization process. It secretes effector proteins affecting other bacterial cells, causing them to die, with further disruption of the natural microbiota [5].

The important determinant of *P. aeruginosa* pathogenesis, especially in persistent infections, is the ability to form a biofilm. This feature is crucial, especially in lung infections in patients with cystic fibrosis, chronic obstructive pulmonary disease, and urinary tract infections (UTI), and in patients with long-term bladder catheter and chronic wounds or infections associated with medical devices or implants. Once established, the biofilm is difficult to eradicate. Bacteria living in the biofilm exhibit higher resistance to antibiotics; thus, the treatment of biofilm-related persistence in *P. aeruginosa* infections is challenging [10].

## 3. *P. aeruginosa* Mechanisms of Resistance

Infections caused by *P. aeruginosa* are difficult to treat, as these microorganisms exhibit high inherent (natural) resistance to many antibiotics (aminopenicillins, first- and second-generation cephalosporins, orally administered and some intravenously administrated third-generation cephalosporins, trimethoprim/sulfamethoxazole, tetracyclines) [11] and the unique ability to develop resistance to almost all antibacterial drugs.

One of the natural mechanisms of antibiotic resistance is the restricted permeability of the outer membrane, which is 12–100 times less permeable than that of *Escherichia coli*. The other mechanisms include the constitutive expression of efflux pumps and the production of antibiotic-inactivating enzymes. Acquired resistance mechanisms result from mutations in chromosomal DNA or from horizontal gene transfer from other bacteria. They extend the natural resistance to antibiotics and lead to the development of multidrug-resistant (MDR) and pandrug-resistant (PDR) strains. Major mechanisms are loss of porins, modification of antibiotic targets, overexpression of efflux pumps, and enzymatic inactivation of antibiotics [12].

From a clinical point of view, the most important resistances in *P. aeruginosa* are against β-lactam antibiotics, aminoglycosides, quinolones, and colistin. The acquired β-lactam resistance is the result of several mechanisms such as mutations causing specific outer membrane protein (OMP) deficiencies (e.g., mutations in OprD, which is the main OMP for uptake of carbapenems in *P. aeruginosa*), the upregulation of the active efflux system, cephalosporinase AmpC overexpression, modification of PBPs (penicillin-binding proteins), or the production of carbapenemases, e.g., metalo-β-lactamases (MBLs) that hydrolyze β-lactams, with the exception of aztreonam. The major acquired mechanisms of resistance to aminoglycosides occurs through changes in the aminoglycoside target (the 30S ribosomal subunit), the production of aminoglycoside-modifying enzymes, as well as overexpression of the efflux pumps. Resistance to quinolones is mainly associated with the overexpression of efflux pumps such as the MexAB-OprM and the mutation in genes encoding for target enzymes (topoisomerase IV and DNA gyrase) [13].

A worrying phenomenon is the emergence of resistance to colistin—a last resort antibiotic in the treatment of infections caused by MDR *P. aeruginosa***.** The main mechanisms of resistance to this antibiotic are chromosomal mutations, leading to the modification of LPS and resulting in a decrease in the affinity of lipid A for polymyxins. Another much more dangerous phenomenon from an epidemiological point of view is the association with the transmissibility of plasmid *mcr*-genes, first described in 2015 in China, and then discovered in several regions of the world in various animals and humans enteric bacteria [14]. Currently, the plasmid *mcr*-genes are also detected in *P. aeruginosa* [15]. In some countries, especially in those where colistin is overused in human and veterinary medicine and is widely used in the poultry industry as a growth promoter, the percentage of *P. aeruginosa* strains resistant to colistin reaches more than 20% [15].

In addition to the high levels of intrinsic and acquired resistance to many clinically used antibiotics, *P. aeruginosa* has the ability to display adaptive resistance based mainly on biofilm formation and the production of multidrug-tolerant persister cells [16].

MDR *P. aeruginosa* strains can be selected de novo due to the selection pressure of antibiotics (pre-existing antibiotics susceptible *P. aeruginosa* when exposed to antibiotics may eventually become resistant) or can be acquired through horizontal transmission, as seen during an epidemic when a single clone was transferred between patients in the same hospital ward [17].

*P. aeruginosa* may use the same mechanism to develop resistance to antibiotics from different classes. For example, the Mex AB-OprM efflux system is active against fluoroquinolones, beta-lactams and aminoglycosides. Thus, exposure to one drug, e.g., fluoroquinolone or β-lactam, may select mutants resistant to other classes that are substrates for the same efflux pump. Another mechanism of multidrug resistance development is the acquisition of genes encoding various resistance mechanisms on mobile genetic elements [18].

## 4. *P. aeruginosa* Colonization

The intestinal colonization of *P. aeruginosa* and the translocation of this bacterium from the gastrointestinal tract into the bloodstream is considered a key in the pathogenesis of some severe endogenous *P. aeruginosa* infections in immunocompromised people, including cancer patients [19]. This is mainly the case of BSI (blood stream infections) in patients with neutropenia, although in lung infections, hematogenous spread or direct contamination of the lungs by *P. aeruginosa* from the intestinal tract is also possible [19]. The colonization usually precedes *P. aeruginosa* hospital infections. It has been demonstrated that in ICU (intensive care unit) patients, the risk of *P. aeruginosa* infection is 15 times higher in colonized patients than in noncolonized ones [20]. Studies by Mendes et al. and Sadowska-Klasa et al. have shown an association between pre-colonization of the gut by *P. aeruginosa* resistant to carbapenems (CRPA) and bloodstream infection caused by the same strains in patients undergoing HSCT (hematopoietic stem cell transplant). The incidence of MDR bacteremia is higher in patients with MDR colonization than in non-colonized patients [21,22].

*P. aeruginosa* is not a common member of the gut microbiota in healthy people. In the non-hospital population, the carrier prevalence of this microorganism was estimated at 1.47% in adults [23] and 5% in children [24]. Community strains show great genetic diversity and general susceptibility to antibiotics [23,24]. In hospitalized patients, the risk of *P. aeruginosa* colonization may increase many times over, especially with the use of wide-ranging active antibiotics. For example, in one study, 43% of patients hospitalized in the ICU (intensive care unit) were colonized with these bacteria [25]. The source of *P. aeruginosa* colonization is the hospital environment where *P. aeruginosa* is ubiquitous. It has been shown that patients treated in the ICU and exposed to antibiotics can be colonized by various *P. aeruginosa* strains (initially mainly by non-MDR, later also by MDR and PDR). The main risk factor for acquiring multi-drug resistant strains is prior exposure to carbapenems, including ertapenem or fluoroquinolones [25].

Over 20 years ago, Andremont et al. assessed the colonization of *P. aeruginosa* in patients hospitalized in the oncohematology department at over 30% [26]. Currently, when anticancer therapy is more aggressive and the antibiotic therapies and prophylaxis in this group of patients are frequent, the rate of colonization can be higher. For example, one study shows that during hospitalization, 74.2% of patients with leukemia, lymphoma, and multiple myeloma developed rectal colonization and 25.8% developed throat colonization by XDR *P. aeruginosa* (extensively drug-resistant *P. aeruginosa*). The risk factors were the usage of ciprofloxacin, as well as more than three different antibiotics during the time of hospitalization, the usage of medical devices such as catheters, and a C-reactive protein >10 mg/dL. The last factor, in the authors’ opinions, may be related to the weakening of the protective mucosal barrier as a result of a pro-inflammatory state or a concomitant infectious disease [27].

Antibiotics disrupt the natural intestinal microbiota and facilitate colonization with pathogenic microorganisms, including *P. aeruginosa* [28]. They not only change the composition of the taxa, they also affect the metabolism, gene expression and protein activity of microorganisms inhabiting the gastrointestinal tract. The complex gut microbiota (both in terms of microbial diversity and density) play a key role in providing resistance to infections. This mechanism is known as “colonization resistance”. Conversely, dysbiosis causes a loss of protection against colonization [29]. In cancer patients, not only antibiotics, but also anticancer chemotherapy alone, causes changes in the oral microbiota depending on the transition from Gram-positive to Gram-negative bacteria, including *Klebsiella*, *Enterobacter*, *Pseudomonas* and *Escherichia* [30]. Cancer patients, especially those with hematological malignances, and hematopoietic stem cell transplant (HSCT) recipients display reduced gut microbiota diversity. The cause of this situation is aggressive chemotherapy and frequent use of antibiotics for treatment and prevention. Dysbiosis, which occurs along with damage to the mucosa, which is common in this group of patients, enables the translocation of pathogens through the damaged intestinal epithelium into the bloodstream and poses potential life-threatening infections, especially during episodes of neutropenia [31].

Moreover, other host factors common in solid tumor patients, such as surgical intervention, severe trauma, and obstruction, may favor *P. aeruginosa* colonization of the intestines. In addition to the intestines, *P. aeruginosa* can also colonize the throat, nose, skin, and urinary tract [32]. Under conditions of severe stress, trauma, and surgery, *P. aeruginosa* may become more virulent and damage the intestinal epithelium, and possibly promote tumor formation in predisposed hosts [33]. While intestinal colonization is considered the most important reservoir of *P. aeruginosa* in endogenous pseudomonal infections in hematology patients, some studies also point to other sources [34]. In addition to the endogenous source of infection with *P. aeruginosa*, exogenous routes are also possible. They mainly play a role during outbreaks, when *P. aeruginosa* comes directly from the contaminated hospital environment or from other infected or colonized patients [20] (Figure 2).

## 5. *P. aeruginosa* as an Etiological Factor of Infections in Cancer Patients

*P. aeruginosa* is one of the most common opportunistic pathogens [35] causing various forms of acute and chronic infections such as bloodstream infections (BSI), chronic lung infections in cystic fibrosis patients, soft skin infections (SSI) including surgical wounds and burns, urinary tract infections (UTIs), ventilator-associated pneumonia (VAP), and other nosocomial infections in immunocompromised and critically ill patients. Community-acquired *P. aeruginosa* infections are much less common. The examples are: hot tub folliculitis, otitis externa (swimmer’s ear) and keratitis in individuals extendedly wearing contact lens [35]. In cancer patients, *P. aeruginosa* causes mainly BSI, respiratory infections, UTI, and skin and soft tissue infections [36]. The most dangerous infection caused by *P. aeruginosa* and occurring in cancer patients is blood stream infection. The risk factors for the development of *P. aeruginosa* BSI in oncological patients are: neutropenia, use of corticosteroids, severity of underlying disease, or prior surgery [37].

The data from the SENTRY Antimicrobial Surveillance Program collecting microorganisms isolated from patients suffering from BSI show that *P. aeruginosa* is the fourth most common BSI etiological agent and is responsible for 5.3% of cases [38]. In cancer patients, especially in those with hematologic malignances, this percentage is higher (11–18% all BSI cases). In this group of patients, *P. aeruginosa* is second or third after *E. coli* and *K. pneumoniae* as a causative agent of BSI [39,40,41,42].

In *P. aeruginosa* BSI in hematological patients, the mortality rate is high [37]. It has changed from 90% in the 1960s, to approximately 70% in the 1970s, to 20% in the 1990s. During this period, anti-pseudomonal antibiotics were systematically introduced and started to be used empirically [43]. Unfortunately, now that MDR *P. aeruginosa* is more frequent than before, the mortality rate is increasing and can be as high as 70% if MDR *P. aeruginosa* is the causative agent [44]. In general, in patients with febrile neutropenia, the mortality in MDR *P. aeruginosa* BSI is several times higher than in patients without MDR *P. aeruginosa* BSI [45,46]. Independent risk factors for mortality in *P. aeruginosa* BSI (bloodstream infections) in neutropenic patients are inappropriate empiric antibiotic therapy, pneumonia and septic shock at onset [47].

The international, large, retrospective, multicenter study performed between January 2006 and May 2018 showed that 25.4% of *P. aeruginosa* blood stream infections in neutropenic cancer patients was caused by multidrug-resistant strains. The risk factors of this etiology were: hematological disease, the presence of a urinary catheter, prior fluoroquinolone prophylaxis, and prior exposure to broad-spectrum antipseudomonal β-lactams such as piperacillin–tazobactam or carbapenem. The identification of patients with a high risk of MDR *P. aeruginosa* BSI allows for the early introduction of appropriate antibiotic therapy and avoids the overuse of broad-spectrum antibiotics where it is not necessary [48]. 

In some neutropenic leukemia patients, *P. aeruginosa* BSI may be secondary to mouth ulcers and necrotic skin lesions (ecthyma gangrenosum) caused by this microorganism. Although oral lesions can be cured as a result of appropriate antibiotic therapy, after a few weeks, some patients develop blood stream infection caused by *P. aeruginosa* resistant to previously used antibiotics [49]. Ecthyma gangrenosum, an uncommon but dangerous complication in neutropenic patients, is usually caused by *P. aeruginosa*. Erythematous skin lesions on the buttocks, legs, axilla or other part of a body rapidly progress into necrotic ulcers with black eschars. This skin infection is usually (~58% of cases) associated with potentially fatal sepsis, with a mortality rate ranging from 38% to 77% [50]. In such a situation, *P. aeruginosa* is isolated from the blood and from skin lesions. Early diagnosis, intensive empirical antibiotic therapy with the *P. aeruginosa* spectrum, and rapid clearing are critical to patient prognosis [50,51].

*P. aeruginosa* can also be involved in polimicrobial infections such as perianal infections, which are relatively common problems in patients with acute leukemia receiving chemotherapy. These infections usually take the form of mild local cellulite or abscesses and fistulas. The main symptoms of local infection are: pain, discomfort swelling, and constipation. Sometimes these local infections can lead to life-threatening sepsis in patients with severe neutropenia [52]. In oncological patients, *P. aeruginosa* can also be present in the blood due to bacteriemic pneumonia (PABP). These infections, potentially fatal, occur mainly in neutropenic patients undergoing chemotherapy, but can also occur in patients with a solid tumor who have developed myelosuppression following chemotherapy. The clinical presentation of classic PABP is primarily fever without respiratory symptoms, followed by rapid progression of pneumonia. In some cases, abscess formation may occur, which is a rare but possible outcome with a poor prognosis. In PABP, *P. aeruginosa* can be isolated from blood and sputum samples [53].

Although *P. aeruginosa* is more frequent in hematological patients, it can also complicate anti-cancer treatment of patients with solid tumors, where prolonged, deep neutropenia is less common than in hematological patients [54]. In this heterogenous group, risk factors of infection are associated with tumor-related disruption of natural anatomic barriers or obstruction as well as with the type of treatment, such as surgery, chemotherapy, radiotherapy, or immunotherapy. Some *P. aeruginosa* infections in patients with solid tumors can be slightly different than those in oncohematology. The examples are: polimicrobial post-obstructive pneumonia caused by bronchial obstruction, UTI due to urinary stasis, or surgical infection after tumor resection [54].

*P. aeruginosa* has been described as the etiological agents of surgical infections associated with breast implantations in women undergoing anti-cancer treatment [55] and in patients after major head and neck oncological surgical procedures. Surgical site infections caused by this microorganism are often associated with an extended patient’s stay in the hospital, worsening of the patient’s condition and quality of life, or a delay in cancer therapy [56].

*P. aeruginosa* together with *Staphylococcus aureus* are the most common organisms isolated from symptomatic skin metastases from primary malignancies such as breast, lung, gastrointestinal, gynecological or high-risk skin cancer metastases [57]. The normal skin microbiota are disturbed and such symptoms such as ulceration, erosion, pain, malodor, or bleeding are visible in these medical situations. Systemic antibiotics often help patients to improve when combined with topical medications [57].

*P. aeruginosa* and *Acinetobacter baumannii* are also common etiological factors of ventilator-associated pneumonia (VAP) in cancer patients admitted to the intensive care unit (ICU) and requiring mechanical ventilation support [58]. In addition, *P. aeruginosa* is now identified as a common etiological factor of bacterial complications (lower respiratory tract infections, bacteremia, urinary tract infections) in cancer patients affected by COVID-19 [59]. Examples of clinical forms of *P. aeruginosa* infections in cancer patients are presented in Figure 3.

## 6. The Old and the New Antibiotics against *P. aeruginosa*

The broad intrinsic and growing acquired resistance of *P. aeruginosa* to antibiotics makes treatment of infections caused by this microorganism more difficult. The range of antibiotics that can be used against *P. aeruginosa* infections is limited, even when the causative agent is a susceptible wild-type strain. The possible therapeutic options are: antipseudomonal β-lactams (e.g., ceftazidime, cefepime, piperacillin/tazobactam, imipenem, meropenem, aztreonam), fluroquinolones (ciprofloxacin, levofloxacin), aminoglycosides (amikacin, tobramycin), polymyxins (colistin), and fosfomycin, as well as a few drugs recently (in the last 10 years) introduced into medical practice [60]. These novel antibiotics are as follows: β-lactam/β-lactamase inhibitors (BL/BLI) such as ceftolozane/tazobactam (C/T), ceftazidime/avibactam (C/A), meropenem/vaborbactam (M/V), imipenem/relabactam (I/R) and cefiderocol as well as aminoglycoside plazomycin [61]. The portfolio of drugs for the treatment of *P. aeruginosa* infections caused by *P. aeruginosa* is presented in Table 1.

C/T consists of a well-known β-lactamase inhibitor and a new cephalosporin, which is the “better version” of ceftazidime. Ceftolozane exhibits reduced affinity for AmpC β-lactamase naturally produced by *P. aeruginosa* and high affinity for penicillin-binding proteins. In addition, it is stable against other resistance mechanisms such as the overexpression of the efflux MexAB-OprM pump or the loss of OprD [62]. In other BL/BLIs (C/A, M/W an I/R), “the old” antipseudomonal β-lactams are combined with novel β-lactamase inhibitors (avibactam, waborbactam and relebactam) being synthetic non-β-lactam compounds. The addition of these inhibitors restores the activity of β-lactam antibiotics against Gram-negative rods producing some β-lactamases, e.g., cefalosporynases AmpC, carbapenemases, KPC and some oxacillinase clase D (OXA)-C/A, I/R but not metalo-β-lactamases (MBL) [63]. Studies show a variable percentage of clinical *P. aeruginosa* isolates susceptible to new BLs/BLI [64,65,66,67]. The susceptibility patterns differ between geographical region and between hospital settings and depend on local epidemiology, antibiotic policy, patient profiles and the composition of examined strain collection [68]. In our last study [69], we assessed the activity of ceftolozane/tazobactam, ceftazidime/avibactam, meropenem/vaborbactam and colistin against 150 *P. aeruginosa* strains isolated from oncological patients. Our collection consisted mainly of strains exhibiting various mechanisms of resistance to carbapenems, including MBL producers. We showed that 87%, 88%, and 58% of tested strains were susceptible to C/T, C/A and M/V, respectively. The best activity was displayed by colistin (99%). Our study shows that C/T and C/A are better choices than M/V for treatment of infections caused by carbapenem-resistant *P. aeruginosa* strains, provided that the resistance is not due to MBL production. In another polish study, the C/T activity against eight *P. aeruginosa* strains not susceptible to carbapenems, isolated from onco-hematological patients, was 100% [70]. Among new β-lactam antibiotics, only cefiderocol has activity against MBL-producers. It is a siderophore cephalosporin entering into a bacterial cell using an active iron-transport system (“Trojan horse” strategy). This unique mechanism of cell entry breaks down resistance mechanisms such as overexpression of efflux pumps and a loss of the porin channel. Additionally, cefiderocol is stable against carbapenemases (KPC, OXA, and MBL) [63]. However, lack of susceptibility to cefiderocol has already emerged, even in strains not previously exposed to this antibiotic [71].

Another potential option, especially for the treatment of infections caused by Gram-negative bacteria producing MBL and ESBL simultaneously, might be the combination of aztreonam with ceftazidime/avibactam [72].

Novel β-lactam antibiotics have a good safety profile and tolerability and predictable pharmacokinetics. The problem is the availability, the high costs of therapy, and the lack of an oral formulation [73]. Incidentally, the only drugs against MDR *P. aeruginosa* infections that can be administered orally are quinolones and fosfomycin (only in UTI infections), which makes it difficult to treat patients with mild infections on an outpatient basis.

Last resort measures against MDR *P. aeruginosa* strains are also old. Previously and rarely used antimicrobials such as colistin and fosfomycin have recently been revitalized. Unfortunately, there are significant limitations in their utility opposed by their toxicity (nephrotoxicity and neurotoxicity of colistin), increasing resistance to these drugs, and worse effectiveness than β-lactams in the treatment of severe infections. Moreover, none of them can be used in monotherapy [74]. In the case of colistin, the risk of nephrotoxicity may be related to the following factors: daily body dose of colistin, low serum albumin, chronic kidney disease, cancer, diabetes, anemia and other kidney diseases [75]. In oncological patients, the nephrotoxicity of this antibiotic may overlap with kidney damage resulting from anti-cancer therapy. Another disadvantage of colistin is the risk of the emergence of resistance during therapy (adaptive resistance). To counteract this and to increase the effectiveness of treatment, high doses of colistin are recommended for the treatment of invasive infections and in combination with one or two other antimicrobial agents to which the organism is susceptible. When no active antibiotic is available, those with the lowest MICs (minimum inhibitory concentrations) should be chosen by interpretation in relation to the breakpoints [76]. For salvage therapy, colistin can be used in combination with aminoglycosides, carbapenems or fosfomycin [77].

Fosfomycin is another old antibiotic with a broad spectrum of activity against G-negative bacilli (except *Acinetobacter* spp.) as well as against Gram-positive bacteria. Until recently, it was used as an oral drug to treat community-acquired urinary tract infections. Intravenous formulations available in many countries are currently used in Gram-negative hospital strains of MDR. As resistance to this drug may develop during therapy, fosfomycin should be used in combination with other antibiotics [78].

## 7. Treatment of *P. aeruginosa* Infections in Cancer Patients

Empirically adequate antibacterial therapy is crucial in the management of severe, life-threatening infections in oncological patients undergoing anti-cancer treatment. Antibiotics with anti-*Pseudomonas* activity should be administrated in these groups of patients where infections of this etiology are highly probable and are often associated with a poor outcome, such as oncohematological febrile neutropenic patients [43]. Treatment success largely depends on the stage of the underlying disease, the response to anti-cancer therapy, the clinical form of infection, as well as the sensitivity of the etiological factor to antibiotics and the time of introducing antibiotic therapy. Antimicrobial resistance and/or inadequacy of empirical antibiotic treatment are associated with poorer outcomes in cancer patients with bloodstream infections due to MDR Gram-negative isolates [79].

Blood stream infections caused by *P. aeruginosa* are quite frequent in neutropenic patients and are considered as life-threatening conditions; thus, antibiotics with antipseudomonal activity for empirical treatment should be administered early, directly after the collection of samples for microbiological assays. Various national and international guidelines [80,81,82] concerning the management of infections in febrile neutropenia recommend the use of an antipseudomonal β-lactam (piperacillin/tazobactam, ceftazidime, cefepime, imipenem, meropenem) with or without aminoglycoside to cover possible *P. aeruginosa* strains. If there a risk of infection caused by MDR strains (e.g., prior carbapenem treatment, MDR *P. aeruginosa* colonization or the high frequency of MDR strains in the patient’s setting), drugs such as colistin or new β-lactam, e.g., ceftolozane/tazobactam or ceftazidime/avibactam, should be taken into consideration [82].

It is not clear whether severe *P. aeruginosa* infections should be treated with a single β-lactam or with a combination: β-lactam and aminoglycoside. Although there are studies showing that both strategies have comparable efficacy [43], it appears that adding an aminoglycoside to an another pseudomonal drug may be beneficial in the initial treatment until susceptibility testing is obtained [83]. This mainly applies to seriously ill patients who stay in the hospital for a long time, where the risk of infection with a multi-drug-resistant strain is high [18].

Although the advantage of combination therapy over monotherapy is unclear, most guidelines, e.g., IDSA (Infectious Diseases Society of America), ESMO (European Society for Medical Oncology), propose the usage of β-lactam with aminoglycoside or fluoroquinolone in the first-line therapy of FN (febrile neutropenia) BSI caused by Gram-negative bacteria. When the results of susceptibility testing are known and the patient is stable, the therapy can be continued only with β-lactam [80,81].

An aminoglycoside added to a β-lactam may cover some resistant strains and may broaden the antimicrobial spectrum, improving the appropriateness of the empirical treatment, which can result in a reduction of mortality, especially in hematological neutropenic patients. In addition, the aminoglycoside used in short cycles and in once-per-day doses does not significantly impair renal function [84].

To improve treatment outcomes in patients with severe *P. aeruginosa* infections, some β-lactams (cefepime, piperacillin/tazobactam, meropenem) may be administered as a prolonged infusion. As demonstrated in vitro and using animal models, this mode of β-lactam administration reduces the emergence of *P. aeruginosa* resistance and increases bactericidal activity compared to standard infusion and ultimately improves patient outcomes according to the observational data [83,85].

The guidelines prepared by The Expert Group of the 4th European Conference on Infections in Leukemia (ECIL) in 2013 also recommend a “de-escalation” strategy in patients with a high risk of infection, including *P. aeruginosa* BSI. This strategy provides a broad spectrum of activity of antibiotics used in empirical therapy that are able to cover even resistant strains and that narrow the spectrum of antibiotics after receiving a microbiological test result [86]. Local resistance patterns, the length of hospital stay, and actual or previous colonization or infection with MDR bacteria, as well as the patient risk factors for a complicated clinical course, are helpful in the individualization of empiric therapy. In some centers where the percentage of MDR Gram-negative rods is high, colistin/polymyxin B and fosfomycin in combination with other antibiotics (e.g., carbapenems, aminoglycosides) should be considered as the first line of therapy. Especially in FN patients, drugs such as fosfomycin or colistin have poorer efficacy, even when they are used in combination with other drugs, than the antibiotics they replace [87,88].

In some patients with a low risk of infection caused by MDR bacteria and a low risk for complications of severe infection, after the first intravenous doses, the oral treatment with combination of, e.g., amoxicillin/clavulanic acid and fluoroquinolone (provided the patient has not received prophylaxis with this drug), could be administrated. Unfortunately, in many oncological centers, the resistance to Gram-negative rods, including *P. aeruginosa*, are high; thus, therapy with fluoroquinolone is not a good option [80,82].

## 8. Prevention of *P.*
*aeruginosa* Infection in Cancer Patients

The high possible exposure to *P. aeruginosa* in a hospital setting and limited possibilities of an effective treatment for MDR *P. aeruginosa* infections highlight the importance of preventive measures such as limiting the transmission of resistant strains, active surveillance screening for MDR colonization, microbiological diagnostics, proper management of intravascular catheters, antimicrobial stewardship and antimicrobial prophylaxis [83]. To limit the transmission of MDR bacteria in oncological wards, universal procedures (e.g., hand hygiene, protection measures—gloves, masks, gowns— and contact isolation from patients infected/colonized by the MDR strain) must be rigorously followed. Patients with the highest risk of infection (e.g., HSCT recipients or people prone to prolonged and profound neutropenia) should be isolated in single-patient rooms with HEPA (high-efficiency particulate air) filtration. In rooms where neutropenic patients are hospitalized, plants and flowers should not be allowed. Sometimes, a low-bacteria diet (cooked meals) is recommended for this group of patients [80,82].

To detect patients colonized by MDR bacteria, microbiological surveillance screening can be carried out. Many hematological/oncological centers perform microbiological screening for selected MDR bacteria at admission. Usually, patients are screened for MRSA (nose swabs) and VRE (rectal swabs) and carbapenem-resistant Enterobacterales (rectal swabs). Patients colonized/infected by the most dangerous resistant strains are isolated in single rooms depending on the availability of individual rooms in the facility [89]. Moreover, knowledge of the colonizing microorganism can be helpful in guiding the empirical antibiotic therapy [21]. Unfortunately, although surveillance stool screening (SSS) can be useful to identify hematological patients (HSCT recipients) with intestinal colonization by MDR *Enterobacterales*, in the case of *P. aeruginosa*, SSS is not highly effective because the intestinal tract is not the only possible source of infection in this group of patients [83]. Moreover, as it was mentioned above, the *P. aeruginosa* is not a typical member of the gut microbiota, and this colonization sometimes is not easy to detect. It seems that routine *P. aeruginosa* screening should primarily be performed during hospital outbreaks and in these hematological wards where MDR *P. aeruginosa* strains are frequently isolated [90]. To increase the accuracy of *P. aeruginosa* carriage detection, the combinations of samples (stool, pharyngeal swab and urine) can be taken and analyzed [91].

In general, the symptoms of *P. aeruginosa* and other Gram-negative infections are non-specific and are impossible to distinguish from infections caused by other bacteria on physical examination. Therefore, microbiological assays are the only way to determine the causative agents and their susceptibility to antibiotics [74]. To detect *P. aeruginosa* in biological samples, traditional cultures and if possible rapid diagnostics based on polymerase chain reaction should be performed. The results of susceptibility testing allow for the administration of an appropriate treatment, e.g., de-escalate initial antibiotic therapy, choose the best antibiotic, and limit the usage of drugs reserved for MDR bacteria if the susceptible strain is detected [83]. Rapid molecular diagnostics, e.g., tests detecting the mechanisms of resistance or, in the near future, new antimicrobial susceptibility testing [92], can be useful in the early optimalization of empirical therapy and more targeted usage of some antibiotics. For example, the rapid detection of MBL enzyme produced by *P. aeruginosa* excludes the usage of drugs such as C/T or C/A.

Another method of prevention of infection caused by Gram-negative rods, including *P. aeruginosa*, in neutropenic patients can be fluoroquinolone prophylaxis (FP). Many institutional guidelines recommend it for patients at high risk of severe and prolonged neutropenia [80,93,94]. IDSA guidelines for the use of antimicrobial agents in neutropenic patients with cancer, published in 2010, recommend considering a prophylaxis with ciprofloxacin or levofloxacin for high-risk but not for low-risk patients (anticipated duration of neutropenia less than 7 days) [80].

Fluoroquinolones have extensively been used for prophylaxis in neutropenic cancer patients since the 1990s, but now in the era of increasing resistance of bacteria to antibiotics, its role is controversial. Some guidelines discourage using it routinely, even in high-risk patients [81]. Although the use of FP decreases the rate of BSI caused by Gram-negative rods, it does not influence overall mortality, and it increases the risk of colonization/infection caused by resistant strains. Thus, the benefits of this prophylaxis are not obvious [95,96]. It has been shown that the fluoroquinolone prophylaxis in hematological patients is associated with bacteremia caused by meropenem-nonsusceptible *P. aeruginosa* strains with oprD mutations and mutations increasing efflux pumps activity [97]. Currently, the effectiveness of fluoroquinolone prophylaxis is diminished by the high percentage of resistant strains among Gram-negative rods colonizing hospitalized patients [98]. It has been shown that its efficacy is reduced when the prevalence of fluoroquinolone resistance among Gram-negative rods exceeds 20% [99]. The prevalence of *P. aeruginosa* strains resistant to these drugs varies among settings, but in oncological centers for adult patients, it is high: about 50% for ciprofloxacin and 40% or even more for levofloxacin [100]. *P. aeruginosa* susceptibility to fluoroquinolones among strains isolated from FN oncohematological children are usually higher (in one study, it was 97.2%) than in adult patients, because the use of these antibiotics is restricted in children due to concerns of adverse skeletal effects [45]. In addition to antibiotic resistance, other disadvantages of antibiotic prophylaxis are the serious side effects of fluoroquinolones, the risk of fungal overgrowth, and *C. difficile* enteritis. [80] Therefore, monitoring these facts is necessary to assess the effectiveness of FP, in particular, in the oncological ward.

## 9. Conclusions

While cancer treatment efficacy and patient prognosis have improved in recent decades, the risk of complications from cancer therapy, including invasive infections, can lead to increased morbidity and mortality. The identification of patients at the highest risk of infection due to MDR strains, including *P. aeruginosa*, is crucial for infection management in cancer patients in the era of increasing resistance to antibiotics. Microbiology testing selects patients who may benefit from a wide range of antibiotics and avoids the use of new drugs of last resort in patients at low risk of developing resistance. Due to microbiological diagnostics, the general guidelines can be adapted to the local epidemiological situation and local susceptibility patterns.

For a resistant pathogen such as *P. aeruginosa,* the optimal use of currently available antibiotics and newly introduced drugs is particularly important for the efficacy of anticancer therapy and for the protection of patients from death due to severe infections. In this context, as emphasized by many experts, it seems particularly important to re-evaluate the validity and effectiveness of the wide use of fluoroquinolone prophylaxes in patients with neutropenia. The diagnostic and therapeutic path in patients with cancer and infection is presented in Figure 4.

## Figures and Tables

**Figure 1 pathogens-11-00679-f001:**
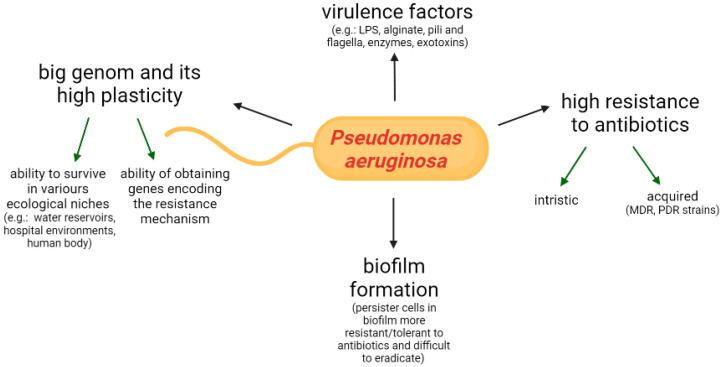
Factors of *Pseudomonas aeruginosa* contributing to its pathogenicity (LPS–lipopolysaccharide, MDR-multidrug-resistant, PDR-pandrug-resistant).

**Figure 2 pathogens-11-00679-f002:**
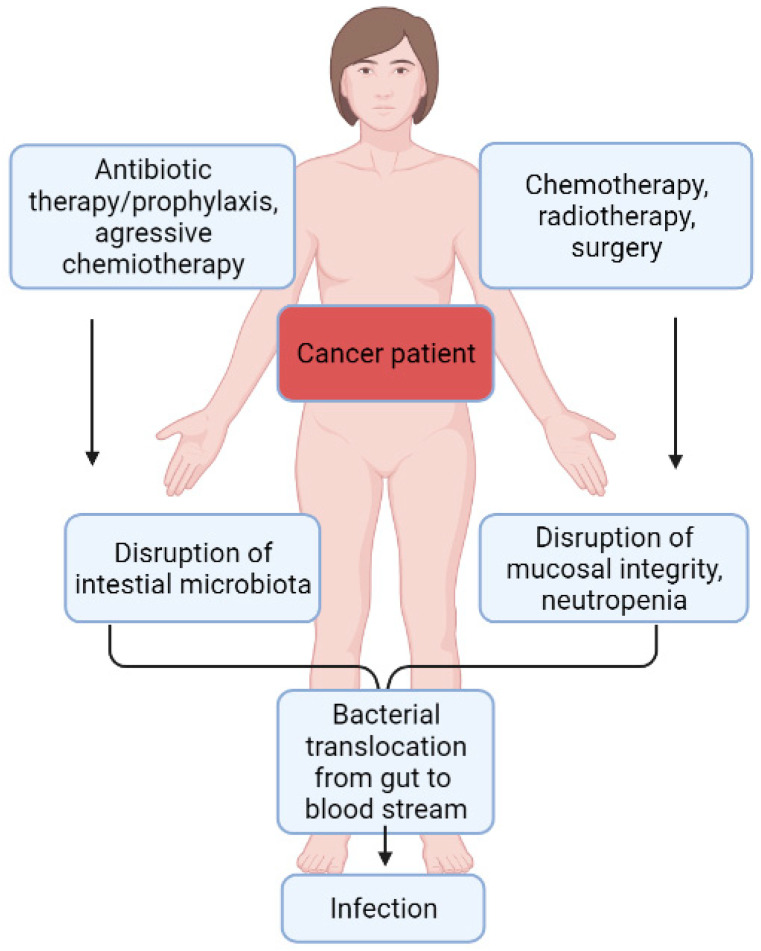
Pathogenesis of endogenous *P. aeruginosa* infections in patients suffering from febrile neutropenia.

**Figure 3 pathogens-11-00679-f003:**
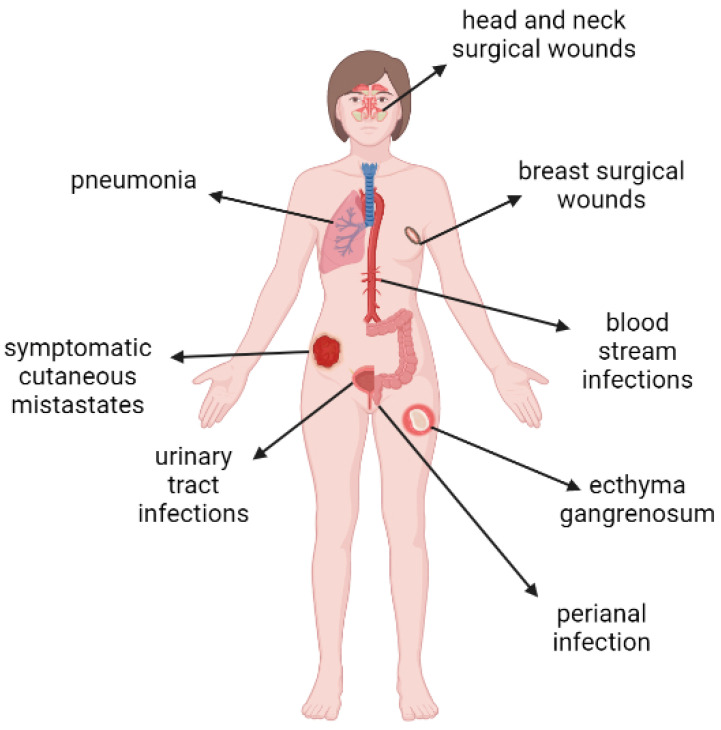
Clinical forms of *P. aeruginosa* infections in cancer patients.

**Figure 4 pathogens-11-00679-f004:**
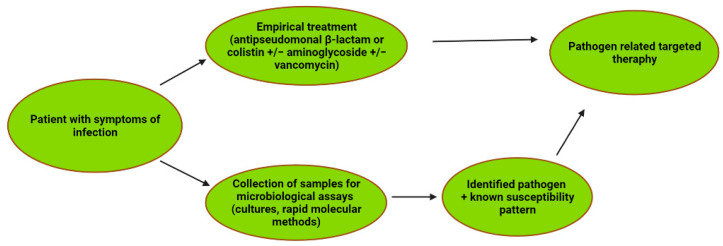
The diagnostic and antibiotic path in cancer patients with infection including *P. aeruginosa* etiology.

**Table 1 pathogens-11-00679-t001:** Antibiotics for the treatment *P. aeruginosa* infections.

Group of Antibiotics	Old	New
**β-lactams**	Ceftazidime	Ceftolozane/tazobactam
	Cefepime	Ceftazidime/avibactam
	Piperacilin/tazobactam	Meropenem/vaborbactam
	Imipenem	Imipenem/relebactam
	Meropenem	Cefiderocol
	Aztreonam	
**Fluoroquinolones**	Ciprofloxacin	-
	Levofloxacin	
**Aminoglycosides**	Amikacin	Plazomycin
	Tobramycin	
**Polomyxins**	Colistin	-
**Phosphonates**	Fosfomycin	-

## Data Availability

Not applicable.

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
