# Peer review of "Pseudomonas aeruginosa Infections in Cancer Patients"

_pathogens, 2022, doi:10.3390/pathogens11060679_

Round 1

Reviewer 1 Report

The manuscript describes, on the assumption, aspects of P. aeruginosa infection in cancer patients. The work is superficial and not followed up by other elements, including the coexistence of P. aeruginosa within the human host. The phenomenon is observed in host-adapted P. aeruginosa strains showing removal of the T3SS. The manuscript does not mention this fact.

The other aspect of the work is editorial. Many words are misspelled (chinolones, polomyxins), not to mention flaunting the use of commas. It would be helpful if the authors sent the manuscript to an editor before submitting it to the journal.

Author Response

RESPONSES TO REVIEWER

Reviewer 1:

Comments and Suggestions for Authors

The manuscript describes, on the assumption, aspects of P. aeruginosa infection in cancer patients. The work is superficial and not followed up by other elements, including the coexistence of P. aeruginosa within the human host. The phenomenon is observed in host-adapted P. aeruginosa strains showing removal of the T3SS. The manuscript does not mention this fact.

To address this issue, we have added a new information describing the T3SS loss in host-adapted P. aeruginosa strains: The type III secretion system (T3SS) is important in the pathogenesis of invasive and acute infections, but most strains of P. aeruginosa that cause chronic infections/colonization (e.g. in cystic fibrosis patients) may inhibit or even lose this and some other virulence factors which result in immune evasion and host-adaptation [7].

The other aspect of the work is editorial. Many words are misspelled (chinolones, polomyxins), not to mention flaunting the use of commas. It would be helpful if the authors sent the manuscript to an editor before submitting it to the journal.

We are sorry for several spelling errors. The text has been modified as necessary.

Reviewer 2 Report

The authors have prepared a well-structured and broad-ranging review of P. aeruginosa infections in cancer patients, ranging from an outline of the mechanisms of pathogenicity and antibiotic resistance through to treatment approaches. The review is generally clearly written though some minor editing is required in some parts and while the authors have reviewed the field extensively, some statements need to be supported by additional references. I also suggest that the authors consider simplifying their title to “Pseudomonas aeruginosa infections in cancer patients”. Comments on specific aspects are given below.

Figure 1. As only one aspect  of the figure deals with virulence factors, this would be better entitled “Factors of Pseudomonas aeruginosa contributing to its pathogenicity”, or something similar.

Some statements made by the authors should be supported by references (or the references should be more clearly linked to the statements). These include the sentences
-       lines 130-131 (mcr plasmid in P aeruginosa)
-       lines 152-155 (translocation of p aeruginosa from GI tract into bloodstream as a key to causing infections)
-       lines 160-163 (Several studies have shown ….)
-       line 217, the cited reference (number 19) is not appropriate for this statement
-       lines 223-227 (P aeruginosa is one of the most common opportunistic pathogens …)
-       lines 271-273 (Mortality rate from erythrematous skin lesions)
-       lines 289 – 294 (infections in patients with solid tumors)
-       lines 304-308 (P aeruginosa and S aureus are the most common organisms from skin metastases)

In addition in some places the authors refer to a single study as “For example” – how typical are these examples, can you cite others?

Line 239 “(over dozen % of all BSI cases)” – does this mean, over 12%?

Lines 251-254 (BSI infections), I did not follow what was meant here.

Lines 276-288 does this whole paragraph relate to polymicrobial infections? Or only the first 4 sentences?

Line 320 (and also the abstract, and Table 1, and elsewhere). What is meant by “Old” and “New” and “Novel”? Does “New” mean, introduced into clinical practice in the last 10 years?

Line 365. In fact, loss of uptake proteins can lead to resistance to cefiderocol as shown in a recent paper (Streling et al, Clin Inf Dis 73: e4472 (2021)

Lines 400 to 414 are treatment-related and would be a better fit in the next section “Treatment of P. aeruginosa infections” than the section on “Old and new antibiotics”

Conclusion lines 555 – 561. It would be nice to have a flow diagram of some sort indicating a pathway of antibiotic use in treating infections in cancer patients  (empirical treatment -> identify pathogen -> pathogen-specific empirical antibiotics -> susceptibility testing -> effective antibiotics) or something like that. There are considerable efforts to use more rapid methods to allow faster diagnostics of species, and antibiotic susceptibility (eg. van Belkum et al Nature Reviews Microbiology 5:299 (2020)) and while these are not yet being used in practice, they likely will be in the future, and the authors could usefully mention this.

Author Response

RESPONSES TO REVIEWER

Reviewer 2:

The authors have prepared a well-structured and broad-ranging review of P. aeruginosa infections in cancer patients, ranging from an outline of the mechanisms of pathogenicity and antibiotic resistance through to treatment approaches. The review is generally clearly written though some minor editing is required in some parts and while the authors have reviewed the field extensively, some statements need to be supported by additional references. I also suggest that the authors consider simplifying their title to “Pseudomonas aeruginosa infections in cancer patients”. Comments on specific aspects are given below.

Thank you for your suggestions, we simplify the title as requested

Figure 1. As only one aspect of the figure deals with virulence factors, this would be better entitled “Factors of Pseudomonas aeruginosa contributing to its pathogenicity”, or something similar.

The title of the Figure 1 has been changed according to your suggestion.  

Some statements made by the authors should be supported by references (or the references should be more clearly linked to the statements). These include the sentences
-lines 130-131 (mcr plasmid in P aeruginosa)

A request modification was made (reference 15th)

-lines 152-155 (translocation of p aeruginosa from GI tract into bloodstream as a key to causing infections)

The information given is contained in the reference 19th

-lines 160-163 (Several studies have shown ….)

This issue was highlighted in the 21st and 22nd reference papers by Mendes et al. and A Sadowska-Klasa et al.

-line 217, the cited reference (number 19) is not appropriate for this statement

The information given is contained in the reference 20th previously 19th

-lines 223-227 (P aeruginosa is one of the most common opportunistic pathogens …)

The information given is contained in the reference 35th

-lines 271-273 (Mortality rate from erythrematous skin lesions)

The information given is contained in the reference 50th

-lines 289 – 294 (infections in patients with solid tumors)

The information given is contained in the reference 54th

-lines 304-308 (P aeruginosa and S aureus are the most common organisms from skin metastases)

The information given is contained in the reference 57th

In addition, in some places the authors refer to a single study as “For example” – how typical are these examples, can you cite others?

Yes, these are just examples.

Line 239 “(over dozen % of all BSI cases)” – does this mean, over 12%?

The text has been modified.

Lines 251-254 (BSI infections), I did not follow what was meant here.

This fragment has been removed.

Lines 276-288 does this whole paragraph relate to polymicrobial infections? Or only the first 4 sentences?

Lines 279-284 relate to polymicrobial infections reference 52nd

Line 320 (and also the abstract, and Table 1, and elsewhere). What is meant by “Old” and “New” and “Novel”? Does “New” mean, introduced into clinical practice in the last 10 years?

This information has been introduced into the current version of the manuscript (line 330)

Line 365. In fact, loss of uptake proteins can lead to resistance to cefiderocol as shown in a recent paper (Streling et al, Clin Inf Dis 73: e4472 (2021)

The information has been completed in the manuscript. (ref. 71)

Lines 400 to 414 are treatment-related and would be a better fit in the next section “Treatment of P. aeruginosa infections” than the section on “Old and new antibiotics”

The text has been modified as requested (lines 428-434, 446-451)

Conclusion lines 555 – 561. It would be nice to have a flow diagram of some sort indicating a pathway of antibiotic use in treating infections in cancer patients  (empirical treatment -> identify pathogen -> pathogen-specific empirical antibiotics -> susceptibility testing -> effective antibiotics) or something like that. There are considerable efforts to use more rapid methods to allow faster diagnostics of species, and antibiotic susceptibility (eg. van Belkum et al Nature Reviews Microbiology 5:299 (2020)) and while these are not yet being used in practice, they likely will be in the future, and the authors could usefully mention this.

Thank you for these points. The diagram (Fig. 4) as well as the information concerning new AST methods and suggested reference has been added to the current version of the manuscript (lines 514-518)

Reviewer 3 Report

In the manuscript ID pathogens-1701891, the authors review and describe the clinical and microbiological features of Pseudomonas aeruginosa infections in cancer patients. Starting from pathogenic characteristics of the microorganism involved in all infectious processes, they focus the attention on cancer-associated infections, underlining the risks factors, the treatment options, guidelines and challenges encountered in this specific category of patients.

The paper is interesting and provides comprehensive information on the clinical point of view of P. aeruginosa cancer-associated infections; from a microbiological point of view, the analysis of antibiotic treatment and its challenges in cancer patients, in relation with the diffusion of drug resistance, is well explained and commented.

However, the manuscript figures should be modified:

-Please correct and provide a more comprehensive Figure 1, adding some examples of each microbiological feature (it is better if the examples are even referred to cancer patient treatment).

-Please unify Figures 2 and 3 into one and provide a further image for other features of the pathogen connected with infections in cancer patients (e.g., mechanisms of antibiotic resistance in cancer patients).

Once solved these minor revisions, the paper can be published in “Pathogens”.

MINOR COMMENTS

-Please after mentioning “Pseudomonas aeruginosa” for the first time always use “P. aeruginosa” throughout the manuscript;

-Lines 17, please correct “one of the most frequent opportunistic microorganisms”;

-Line 21, please correct “which damage host cells, evade host immune

-Line 64, please correct “pili and flagella”;

-Line 111, please rephrase “From the clinical point of view, the most important resistances in P. aeruginosa are against”;

-Line 117, please correct “metalo-β-lactamases (MBLs) which”;

-Line 159, please define here the acronym “ICU”;

-Line 163, please define here the acronym “HSCT”;

-Line 242, please correct “It has been changed”;

-Line 266, please correct “the oral lesion can be cured”;

-Line 332, please correct “The list of drugs for the treatment of P. aeruginosa infections“;

-Line 338, please correct “AmpC”;

-Line 341, please correct “the efflux MexAB-OprM pump”;

-Line 356, please correct “The best activity was displayed by colistin”;

-Line 357, please correct “are a better choice than M/V”;

-Line 390, please correct “those with the lowest MICs should be chosen”;

-Line 443, please define the “FN” acronym;

-Line 545, please type “C. difficile” in italic.

Author Response

RESPONSES TO REVIEWER

Reviewer 3:

In the manuscript ID pathogens-1701891, the authors review and describe the clinical and microbiological features of Pseudomonas aeruginosa infections in cancer patients. Starting from pathogenic characteristics of the microorganism involved in all infectious processes, they focus the attention on cancer-associated infections, underlining the risks factors, the treatment options, guidelines and challenges encountered in this specific category of patients.

The paper is interesting and provides comprehensive information on the clinical point of view of P. aeruginosa cancer-associated infections; from a microbiological point of view, the analysis of antibiotic treatment and its challenges in cancer patients, in relation with the diffusion of drug resistance, is well explained and commented.

 Thank you for your positive comments.

However, the manuscript figures should be modified:

-Please correct and provide a more comprehensive Figure 1, adding some examples of each microbiological feature (it is better if the examples are even referred to cancer patient treatment).

Figure 1 has been modified.

-Please unify Figures 2 and 3 into one and provide a further image for other features of the pathogen connected with infections in cancer patients (e.g., mechanisms of antibiotic resistance in cancer patients).

Thank you for your opinion, but in our opinion, these are Figures on two issues.

Once solved these minor revisions, the paper can be published in “Pathogens”.

MINOR COMMENTS

-Please after mentioning “Pseudomonas aeruginosa” for the first time always use “P. aeruginosa” throughout the manuscript;

The text has been modified as requested

-Lines 17, please correct “one of the most frequent opportunistic microorganisms”;

As requested, the text has been modified

-Line 21, please correct “which damage host cells, evade host immune

The text has been modified as requested

-Line 64, please correct “pili and flagella”;

As requested, the text has been modified

-Line 111, please rephrase “From the clinical point of view, the most important resistances in P. aeruginosa are against”;

The text has been modified as requested

-Line 117, please correct “metalo-β-lactamases (MBLs) which”;

As requested, the text has been modified

-Line 159, please define here the acronym “ICU”;

The text has been modified as requested

-Line 163, please define here the acronym “HSCT”;

As requested, the text has been modified

-Line 242, please correct “It has been changed”;

The text has been modified as requested

-Line 266, please correct “the oral lesion can be cured”;

As requested, the text has been modified

-Line 332, please correct “The list of drugs for the treatment of P. aeruginosa infections“;

The text has been modified as requested

-Line 338, please correct “AmpC”;

As requested, the text has been modified

-Line 341, please correct “the efflux MexAB-OprM pump”;

The text has been modified as requested

-Line 356, please correct “The best activity was displayed by colistin”;

As requested, the text has been modified

-Line 357, please correct “are a better choice than M/V”;

The text has been modified as requested

-Line 390, please correct “those with the lowest MICs should be chosen”;

As requested, the text has been modified

-Line 443, please define the “FN” acronym;

The text has been modified as requested

-Line 545, please type “C. difficile” in italic.

As requested, the text has been modified

Round 2

Reviewer 1 Report

The manuscript has not been changed except for some additions. There are still typographical errors (lines 114,115, 128) that must be corrected. 

The manuscript is a rehash of old parts with the hope that it will pass.

Author Response

RESPONSES TO REVIEWER

Reviewer 1:

The manuscript has not been changed except for some additions. There are still typographical errors (lines 114,115, 128) that must be corrected. 

Lines 114-115 -These corrections were introduced at the request of the Reviewer 3

Line 128- The sentence concerns the emergence of resistance to colistin - an antibiotic of last resort. - there were no previous comments on this line.

The manuscript is a rehash of old parts with the hope that it will pass.

Text fragments were transferred between chapters in accordance with the Reviewer's suggestion no. 2, with which we fully agree. We feel awkward because we are not sure what changes we should make to the text of the work to respond satisfactorily to criticism of the superficiality of our work. We would like to emphasize that the publication is not our goal, our goal is to present the practical experience of over 30 years of work in an oncology center, during which we encounter hundreds of cases of infections caused by Pseudomonas aeruginosa.